# NEURAL ARCHITECTURE DESIGN AND ROBUSTNESS: A DATASET

**Steffen Jung[1,2,\*], Jovita Lukasik[1\*], Margret Keuper[1,2]**
[1] Max Planck Institute for Informatics, Saarland Informatics Campus
`{steffen.jung,jlukasik,keuper}@mpi-inf.mpg.de`
[2] University of Siegen

## ABSTRACT

Deep learning models have proven to be successful in a wide range of machine learning tasks. Yet, they are often highly sensitive to perturbations on the input data which can lead to incorrect decisions with high confidence, hampering their deployment for practical use-cases. Thus, finding architectures that are (more) robust against perturbations has received much attention in recent years. Just like the search for well-performing architectures in terms of clean accuracy, this usually involves a tedious trial-and-error process with one additional challenge: the evaluation of a network's robustness is significantly more expensive than its evaluation for clean accuracy. Thus, the aim of this paper is to facilitate better streamlined research on architectural design choices with respect to their impact on robustness as well as, for example, the evaluation of surrogate measures for robustness. We therefore borrow one of the most commonly considered search spaces for neural architecture search for image classification, NAS-Bench-201, which contains a manageable size of $6\,466$ non-isomorphic network designs. We evaluate all these networks on a range of common adversarial attacks and corruption types and introduce a database on neural architecture design and robustness evaluations. We further present three exemplary use cases of this dataset, in which we (i) benchmark robustness measurements based on Jacobian and Hessian matrices for their robustness predictability, (ii) perform neural architecture search on robust accuracies, and (iii) provide an initial analysis of how architectural design choices affect robustness. We find that carefully crafting the topology of a network can have substantial impact on its robustness, where networks with the same parameter count range in mean adversarial robust accuracy from $20\% - 41\%$. Code and data is available at `http://robustness.vision/`.

## 1 INTRODUCTION

One factor in the ever-improving performance of deep neural networks is based on innovations in architecture design. The starting point was the unprecedented result of AlexNet (Krizhevsky et al., 2012) on the visual recognition challenge ImageNet (Deng et al., 2009). Since then, the goal is to find better performing models, surpassing human performance. However, human design of new better performing architectures requires a huge amount of trial-and-error and a good intuition, such that the automated search for new architectures (NAS) receives rapid and growing interest (Zoph & Le, 2017; Real et al., 2017; Ying et al., 2019; Dong & Yang, 2020). The release of tabular benchmarks (Ying et al., 2019; Dong & Yang, 2020) led to a research change; new NAS methods can be evaluated in a transparent and reproducible manner for better comparison.

The rapid growth in NAS research with the main focus on finding new architecture designs with ever-better performance is recently accompanied by the search for architectures that are robust against adversarial attacks and corruptions. This is important, since image classification networks can be easily fooled by adversarial attacks crafted by already light perturbations on the image data, which are invisible for humans. This leads to false predictions of the neural network with high confidence.

Robustness in NAS research combines the objective of high performing and robust architectures (Dong & Yang, 2019; Devaguptapu et al., 2021; Dong et al., 2020a; Hosseini et al., 2021; Mok

et al., 2021). However, there was no attempt so far to evaluate a full search space on robustness, but rather architectures in the wild. This paper is a first step towards closing this gap. We are the first to introduce a robustness dataset based on evaluating a *complete* NAS search space, such as to allow benchmarking neural architecture search approaches for the robustness of the found architectures. This will facilitate better streamlined research on neural architecture design choices and their robustness. We evaluate all 6 466 unique pretrained architectures from the NAS-Bench-201 benchmark (Dong & Yang, 2020) on common adversarial attacks (Goodfellow et al., 2015; Kurakin et al., 2017; Croce & Hein, 2020) and corruption types (Hendrycks & Dietterich, 2019). We thereby follow the argumentation in NAS research that employing one common training scheme for the entire search space will allow for comparability between architectures. Having the combination of pretrained models and the evaluation results in our dataset at hand, we further provide the evaluation of common training-free robustness measurements, such as the Frobenius norm of the Jacobian matrix (Hoffman et al., 2019) and the largest eigenvalue of the Hessian matrix (Zhao et al., 2020), on the full architecture search space and use these measurements as a method to find the supposedly most robust architecture. To prove the promise of our dataset to promote research in neural architecture search for robust models we perform several common NAS algorithms on the clean as well as on the robust accuracy of different image classification tasks. Additionally, we conduct a first analysis of how certain architectural design choices affect robustness with the potential of doubling the robustness of networks with the same number of parameters. This is only possible, since we evaluate the whole search space of NAS-Bench-201 (Dong & Yang, 2020), enabling us to investigate the effect of small architectural changes. To our knowledge we are the first paper to introduce a robustness dataset covering a full (widely used) search space allowing to track the outcome of fine-grained architectural changes. In summary we make the following contributions:

- We present the first robustness dataset evaluating a complete NAS architectural search space.
- We present different use cases for this dataset; from training-free measurements for robustness to neural architecture search.
- Lastly, our dataset shows that a model's robustness against corruptions and adversarial attacks is highly sensitive towards the architectural design, and carefully crafting architectures can substantially improve their robustness.

## 2 RELATED WORK

**Common Corruptions** While neural architectures achieve results in image classification that supersede human performance (He et al., 2015), common corruptions such as Gaussian noise or blur can cause this performance to degrade substantially (Dodge & Karam, 2017). For this reason, Hendrycks & Dietterich (2019) propose a benchmark that enables researchers to evaluate their network design on several common corruption types.

**Adversarial Attacks** Szegedy et al. (2014) showed that image classification networks can be fooled by crafting image perturbations, so called adversarial attacks, that maximize the networks' prediction towards a class different to the image label. Surprisingly, these perturbations can be small enough such that they are not visible to the human eye. One of the first adversarial attacks, called fast gradient sign method (FGSM) (Goodfellow et al., 2015), tries to flip the label of an image in a single perturbation step of limited size. This is achieved by maximizing the loss of the network and requires access to its gradients. Later gradient-based methods, like projected gradient descent (PGD) (Kurakin et al., 2017), iteratively perturb the image in multiple gradient steps. To evaluate robustness in a structured manner, Croce & Hein (2020) propose an ensemble of different attacks, including an adaptive version of PGD (APGD) (Croce & Hein, 2020) and a blackbox attack called Square Attack (Andriushchenko et al., 2020) that has no access to network gradients. Croce et al. (2021) conclude the next step in robustness research by providing an adversarial robustness benchmark, RobustBench, tracking state-of-the-art models in adversarial robustness.

**NAS** Neural Architecture Search (NAS) is an optimization problem with the objective to find an optimal combination of operations in a predefined, constrained *search space*. Early NAS approaches differ by their *search strategy* within the constraint search space. Common NAS strategies are evolutionary methods (Real et al., 2017; 2019), reinforcement learning (RL) (Zoph & Le, 2017; Li et al., 2018), random search (Bergstra & Bengio, 2012; Li & Talwalkar, 2019), local search (White et al., 2021b), and Bayesian optimization (BO) (Kandasamy et al., 2018; Ru et al., 2021; White et al.,

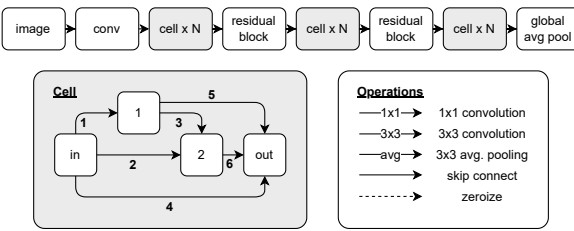

Figure 1: **(top)** Macro architecture. Gray highlighted cells differ between architectures, while the other components stay fixed. **(bottom)** Cell structure and the set of possible, predefined operations. (Figure adapted from (Dong & Yang, 2020))

2021a). Recently, several NAS approaches use generative models to search within a continuous latent space of architectures (Lukasik et al., 2021; Rezaei et al., 2021; Lukasik et al., 2022). To further improve the search strategy efficiency, the research focus shift from discrete optimization methods to faster differentiable search methods, using weight-sharing approaches (Pham et al., 2018; Liu et al., 2019; Bender et al., 2018; Cai et al., 2019; Xie et al., 2019b; Zela et al., 2020). In order to compare NAS approaches properly, NAS benchmarks were introduced and opened the path for fast evaluations. The tabular benchmarks NAS-Bench-101 (Ying et al., 2019) and NAS-Bench-201 (Dong & Yang, 2020) provide exhaustive evaluations of performances and metrics within their predefined search space on image classification tasks. TransNAS-Bench-101 (Duan et al., 2021) introduces a benchmark containing performance and metric information across different vision tasks. We will give a more detailed overview about the NAS-Bench-201 (Dong & Yang, 2020) benchmark in subsection 3.1.

**Robustness in NAS** With the increasing interest in NAS in general, the aspect of robustness of the optimized architectures has become more and more relevant. Devaguptapu et al. (2021) provide a large-scale study that investigates how robust architectures, found by several NAS methods such as (Liu et al., 2019; Cai et al., 2019; Xu et al., 2020), are against several adversarial attacks. They show that these architectures are vulnerable to various different adversarial attacks. Guo et al. (2020) first search directly for a robust neural architecture using one-shot NAS and discover a family of robust architectures. Dong et al. (2020a) constrain the architectures' parameters within a supernet to reduce the Lipschitz constant and therefore increase the resulting networks' robustness. Few prior works such as (Carlini et al., 2019; Xie et al., 2019a; Pang et al., 2021; Xie et al., 2020) propose more in-depth statistical analyses. In particular, Su et al. (2018) evaluate 18 ImageNet models with respect to their adversarial robustness. Ling et al. (2019); Dong et al. (2020b) provide platforms to evaluate adversarial attacks. Tang et al. (2021) provide a robustness investigation benchmark based on different architectures and training techniques on ImageNet. Recently a new line of differentiable robust NAS arose, namely including differentiable network measurements to the one-shot loss target to increase the robustness (Hosseini et al., 2021; Mok et al., 2021). Hosseini et al. (2021) define two differentiable metrics to measure the robustness of the architecture, certified lower bound and Jacobian norm bound, and searches for architectures by maximizing these metrics, respectively. Mok et al. (2021) propose a search algorithm using the intrinsic robustness of a neural network being represented by the smoothness of the network's input loss landscape, i.e. the Hessian matrix.

## 3 DATASET GENERATION

### 3.1 ARCHITECTURES IN NAS-BENCH-201

NAS-Bench-201 (Dong & Yang, 2020) is a cell-based architecture search space. Each cell has in total $4$ nodes and $6$ edges. The nodes in this search space correspond to the architecture's feature maps and the edges represent the architectures operation, which are chosen from the operation set $\mathcal{O} = \{1 \times 1 \text{ conv.}, 3 \times 3 \text{ conv.}, 3 \times 3 \text{ avg. pooling}, \text{skip}, \text{zero}\}$ (see Figure 1). This search space contains in total $5^6 = 15\,625$ architectures, from which only $6\,466$ are unique, since the operations skip and zero can cause isomorphic cells (see Figure 8, appendix), where the latter operation zero stands for dropping the edge. Each architecture is trained on three different image datasets for 200 epochs: CIFAR-10 (Krizhevsky, 2009), CIFAR-100 (Krizhevsky, 2009) and ImageNet16-120

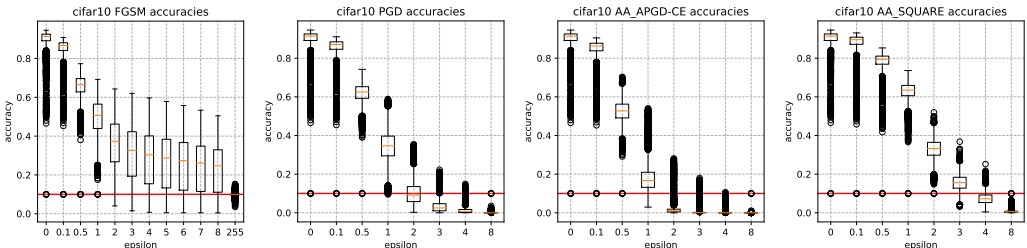

Figure 2: Accuracy boxplots over all 6 466 unique architectures in NAS-Bench-201 for different adversarial attacks (FGSM (Goodfellow et al., 2015), PGD (Kurakin et al., 2017), APGD (Croce & Hein, 2020), Square (Andriushchenko et al., 2020)) and perturbation magnitude values $\epsilon$, evaluated on CIFAR-10. Red line corresponds to guessing. The large spread indicates towards architectural influence on robust performance.

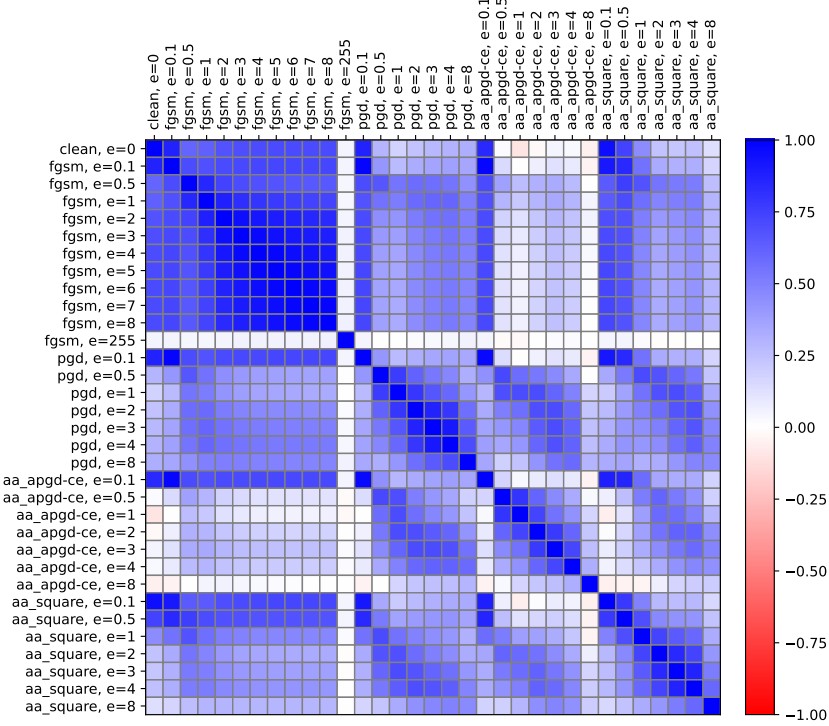

Figure 3: Kendall rank correlation coefficient between clean accuracies and robust accuracies on different attacks and magnitude values $\epsilon$ on CIFAR-10 for all unique architectures in NAS-Bench-201. There seem to be architectural distinctions for susceptibility to different attacks.

(Chrabaszcz et al., 2017). For our evaluations, we consider all unique architectures in the search space and test splits of the corresponding datasets. Hence, we evaluate $3 \cdot 6\,466 = 19\,398$ pretrained networks in total. In the following, we describe which evaluations we collect.

## 3.2 ROBUSTNESS TO ADVERSARIAL ATTACKS

We start by collecting evaluations on different adversarial attacks, namely FGSM, PGD, APGD, and Square Attack. Following, we describe each attack and the collection of their results in more detail.

**FGSM** FGSM (Goodfellow et al., 2015) finds adversarial examples via

$$\tilde{x} = x + \epsilon \text{sign}(\Delta_x J(\theta, x, y)), \tag{1}$$

where $\tilde{x}$ is the adversarial example, $x$ is the input image, $y$ the corresponding label, $\epsilon$ the magnitude of the perturbation, and $\theta$ the network parameters. $J(\theta, x, y)$ is the loss function used to train the attacked network. In the case of architectures trained for NAS-Bench-201, this is cross entropy (CE). Since attacks via FGSM can be evaluated fairly efficiently, we evaluate all architectures for $\epsilon \in E_{FGSM} = \{.1, .5, 1, 2, \ldots, 8, 255\}/255$, so for a total of $|E_{FGSM}| = 11$ times for each architecture. We use Foolbox (Rauber et al., 2017) to perform the attacks, and collect (a) accuracy, (b) average prediction confidences, as well as (c) confusion matrices for each network and $\epsilon$ combination.

**PGD** While FGSM perturbs the image in a single step of size $\epsilon$, PGD (Kurakin et al., 2017) iteratively perturbs the image via

$$\tilde{x}_{n+1} = \mathrm{clip}_{\epsilon, x}(\tilde{x}_n - \alpha \mathrm{sign}(\Delta_x J(\theta, \tilde{x}_n, \tilde{y}))), \tilde{x}_0 = x, \qquad (2)$$

where $\tilde{y}$ is the least likely predicted class of the network, and $\mathrm{clip}_{\epsilon, x}(\cdot)$ is a function clipping to range $[x - \epsilon, x + \epsilon]$. Due to its iterative nature, PGD is more efficient in finding adversarial examples, but requires more computation time. Therefore, we find it sufficient to evaluate PGD for $\epsilon \in E_{PGD} = \{.1, .5, 1, 2, 3, 4, 8\}/255$, so for a total of $|E_{PGD}| = 7$ times for each architecture. As for FGSM, we use Foolbox (Rauber et al., 2017) to perform the attacks using their $L_\infty$ PGD implementation and keep the default settings, which are $\alpha = 0.01/0.3$ for 40 attack iterations. We collect (a) accuracy, (b) average prediction confidences, and (c) confusion matrices for each network and $\epsilon$ combination.

**APGD** AutoAttack (Croce & Hein, 2020) offers an adaptive version of PGD that reduces its step size over time without the need for hyperparameters. We perform this attack using the $L_\infty$ implementation provided by (Croce & Hein, 2020) on CE and choose $E_{APGD} = E_{PGD}$. We kept the default number of attack iterations that is 100. We collect (a) accuracy, (b) average prediction confidences, and (c) confusion matrices for each network and $\epsilon$ combination.

**Square Attack** In contrast to the before-mentioned attacks, Square Attack is a blackbox attack that has no access to the networks' gradients. It solves the following optimization problem using random search:

$$\min_{\tilde{x}} \{ f_{y,\theta}(\tilde{x}) - \max_{k \neq y} f_{k,\theta}(\tilde{x}) \}, \text{ s.t. } \|\tilde{x} - x\|_p \leq \epsilon, \qquad (3)$$

where $f_{k,\theta}(\cdot)$ are the network predictions for class $k$ given an image. We perform this attack using the $L_\infty$ implementation provided by (Croce & Hein, 2020) and choose $E_{Square} = E_{PGD}$. We kept the default number of search iterations at $5\,000$. We collect (a) accuracy, (b) average prediction confidences, and (c) confusion matrices for each network and $\epsilon$ combination.

**Summary** Figure 2 shows aggregated evaluation results on the before-mentioned attacks on CIFAR-10 w.r.t. accuracy. Growing gaps between mean and max accuracies indicate that the architecture has an impact on robust performances. Figure 3 depicts the correlation of ranking all architectures based on different attack scenarios. While there is larger correlation within the same adversarial attack and different values of $\epsilon$, there seem to be architectural distinctions for susceptibility to different attacks.

### 3.3 ROBUSTNESS TO COMMON CORRUPTIONS

To evaluate all unique NAS-Bench-201 Dong & Yang (2020) architectures on common corruptions, we evaluate them on the benchmark data provided by (Hendrycks & Dietterich, 2019). Two datasets are available: CIFAR10-C, which is a corrupted version of CIFAR-10 and CIFAR-100-C, which is a corrupted version of CIFAR-100. Both datasets are perturbed with a total of 15 corruptions at 5 severity levels (see Figure 18 in the Appendix for an example). The training procedure of NAS-Bench-201 only augments the training data with random flipping and random cropping. Hence, no influence should be expected of the training augmentation pipeline on the performance of the networks to those corruptions. We evaluate each of the $15 \cdot 5 = 75$ datasets individually for each network and collect (a) accuracy, (b) average prediction confidences, and (c) confusion matrices.

**Summary** Figure 4 depicts mean accuracies for different corruptions at increasing severity levels. Similar to Figure 2, a growing gap between mean and max accuracies for most of the corruptions can be observed, which indicates towards architectural influences on robustness to common corruptions. Figure 5 depicts the ranking correlation for all architectures between clean and corrupted accuracies. Ranking architectures based on accuracy on different kinds of corruption is mostly uncorrelated. This indicates a high diversity of sensitivity to different kinds of corruption based on architectural design.

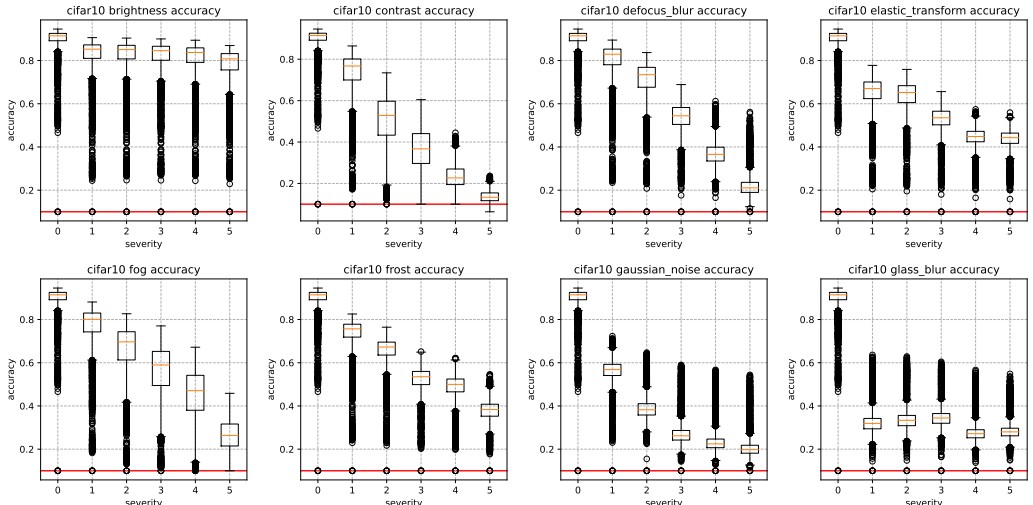

Figure 4: Accuracy boxplots over all unique architectures in NAS-Bench-201 for different corruption types at different severity levels, evaluated on CIFAR-10-C. Red line corresponds to guessing. All corruptions can be found in Figure 21. The large spread indicates towards architectural influence on robust performance.

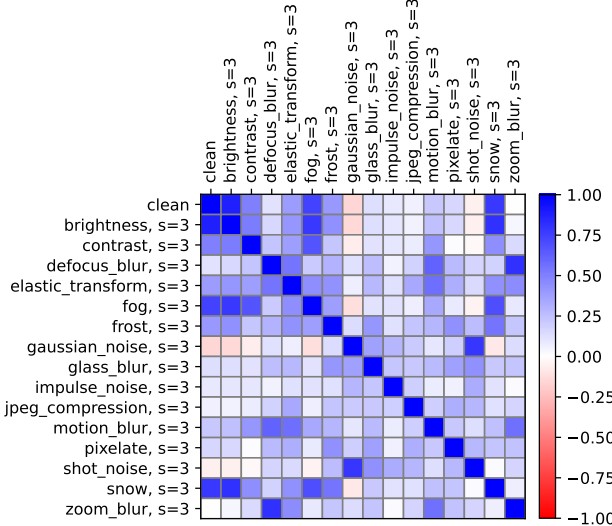

Figure 5: Kendall rank correlation coefficient between clean accuracies and accuracies on different corruptions at severity level 3 on CIFAR-10-C for all unique architectures in NAS-Bench-201. The mostly uncorrelated ranking indicates towards high diversity of sensitivity to different kinds of corruption based on architectural design.

## 4 USE CASES

### 4.1 TRAINING-FREE MEASUREMENTS FOR ROBUSTNESS

Recently, a new research focus in differentiable NAS shifted towards finding not only high-scoring architectures but also finding adversarially robust architectures against several adversarial attacks (Hosseini et al., 2021; Mok et al., 2021) using training characteristics of neural networks. On the one hand, (Hosseini et al., 2021) uses Jacobian-based differentiable metrics to measure robustness. On the other hand, (Mok et al., 2021) improves the search for robust architectures by including the

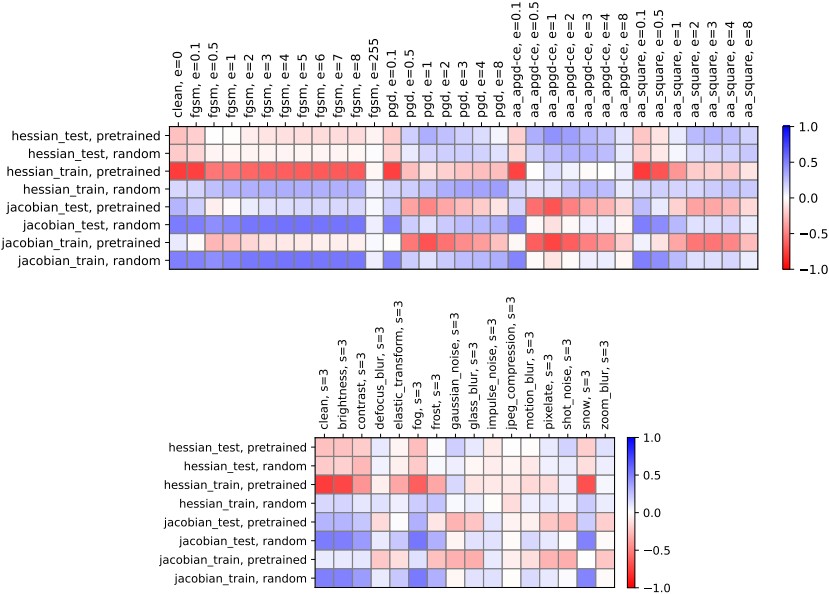

Figure 6: Kendall rank correlation coefficient between Jacobian- and Hessian-based robustness measurements computed on all unique NAS-Bench-201 architectures to corresponding rankings given by **(top)** different adversarial attacks and **(bottom)** different common corruptions. Measurements and accuracies are computed on CIFAR-10 / CIFAR-10-C. Measurements are computed on randomly initialized and pretrained networks contained in NAS-Bench-201. Jacobian-based and Hessian-based measurements correlate well for smaller $\epsilon$ values, but not for larger $\epsilon$ values.

smoothness of the loss landscape of a neural network. In this section, we evaluate these training-free gradient-based measurements with our dataset.

**Background: Jacobian** To improve the robustness of neural architectures, (Hoffman et al., 2019) introduced an efficient Jacobian regularization method with the goal to minimize the network's output change in case of perturbed input data, by minimizing the Frobenius norm of the network's Jacobian matrix, $\mathcal{J}$. The Frobenius norm is defined as $\|J(x)\|_F = \sqrt{\sum_{\phi,c} |J_{\phi,c}(x)|}$. Let $f_\theta : \mathbb{R}^D \to \mathbb{R}^C$ be a neural network with weights denoted by $\theta$ and let $x \in \mathbb{R}^D$ be the input data and $z \in \mathbb{R}^C$ be the output score. Let $\tilde{x} = x + \varepsilon$ be a perturbed input, with $\varepsilon \in \mathbb{R}^D$ being a perturbation vector. The output of the neural network shifts then to $f_{\theta,c}(x + \varepsilon) - f_{\theta,c}(x)$. The input-output Jacobian matrix can be used as a measurement for the networks stability against input perturbations (Hoffman et al., 2019):

$$f_{\theta,c}(x + \varepsilon) - f_{\theta,c}(x) \approx \sum_{d=1}^{D} \varepsilon_d \cdot \frac{\partial f_{\theta,c;d}}{\partial x_d}(x) = \sum_{d=1}^{D} \mathcal{J}_{\theta,c;d}(x) \cdot \varepsilon_d, \qquad (4)$$

according to Taylor-expansion. From Equation 4, we can directly see that the larger the Jacobian components, the larger is the output change and thus the more unstable is the neural network against perturbed input data. In order to increase the stability of the network, (Hoffman et al., 2019) proposes to decrease the Jacobian components by minimizing the square of the Frobenius norm of the Jacobian. Following (Hosseini et al., 2021), we use the efficient algorithm presented in (Hoffman et al., 2019) to compute the Frobenius norm based on random projection for each neural network in the NAS-Bench-201 (Dong & Yang, 2020) benchmark.

**Benchmarking Results: Jacobian** The smaller the Frobenius norm of the Jacobian of a network, the more robust the network is supposed to be. Our dataset allows for a direct evaluation of this statement on all $6\,466$ unique architectures. We use $10$ mini-batches of size $256$ of the training as well as test dataset for both randomly initialized and pretrained networks and compute the mean Frobenius norm. The results in terms of ranking correlation to adversarial robustness is shown in Figure 6 **(top)**, and in terms of ranking correlation to robustness towards common corruptions in Figure 6 **(bottom)**. We can observe that the Jacobian-based measurement correlates well with rankings after attacks by FGSM

and smaller $\epsilon$ values for other attacks. However, this is not true anymore when $\epsilon$ increases, especially in the case of APGD.

**Background: Hessian** Zhao et al. (2020) investigate the loss landscape of a regular neural network and robust neural network against adversarial attacks. Let $\mathcal{L}(f_\theta(x))$ denote the standard classification loss of a neural network $f_\theta$ for clean input data $x \in \mathbb{R}^D$ and $\mathcal{L}(f_\theta(x+\varepsilon))$ be the adversarial loss with perturbed input data $x + \varepsilon, \varepsilon \in \mathbb{R}^D$. Zhao et al. (2020) provide theoretical justification that the latter adversarial loss is highly correlated with the largest eigenvalue of the input Hessian matrix $H(x)$ of the clean input data $x$, denoted by $\lambda_{\max}$. Therefore the eigenspectrum of the Hessian matrix of the regular network can be used for quantifying the robustness: large Hessian spectrum implies a sharp minimum resulting in a more vulnerable neural network against adversarial attacks. Whereas in the case of a neural network with small Hessian spectrum, implying a flat minimum, more perturbation on the input is needed to leave the minimum. We make use of (Chatzimichailidis et al., 2019) to compute the largest eigenvalue $\lambda_{\max}$ for each neural network in the NAS-Bench-201 (Dong & Yang, 2020) benchmark.

**Benchmarking Results: Hessian** For this measurement, we calculate the largest eigenvalues of all unique architectures using the Hessian approximation in (Chatzimichailidis et al., 2019). We use 10 mini-batches of size 256 of the training as well as test dataset for both randomly initialized and pretrained networks and compute the mean largest eigenvalue. These results are also shown in Figure 6. We can observe that the Hessian-based measurement behaves similarly to the Jacobian-based measurement.

Table 1: Neural Architecture Search on the clean test accuracy and the FGSM ($\epsilon = 1$) robust test accuracy for different state of the art methods on CIFAR-10 in the NAS-Bench-201 (Dong & Yang, 2020) search space (mean over 100 runs). Results are the mean accuracies of the best architectures found on different adversarial attacks and the mean accuracy over all corruptions and severity levels in CIFAR-10-C.

| | Method | Test Accuracy ($\epsilon = 1.0$) | | | | | Clean |
| | | Clean | FGSM | PDG | APGD | Squares | |
| | | CIFAR-10 | | | | | CF-10-C |
| | **Optimum** | 94.68 | 69.24 | 58.85 | 54.02 | 73.61 | 58.55 |
| Clean | BANANAS (White et al., 2021a) | 94.21 | 64.25 | 41.10 | 18.62 | 68.69 | 55.52 |
| | Local Search (White et al., 2021b) | 94.65 | 63.95 | 41.17 | 18.74 | 69.59 | 56.90 |
| | Random Search (Li & Talwalkar, 2019) | 94.22 | 63.38 | 40.09 | 17.84 | 68.40 | 55.60 |
| | Regularized Evolution (Real et al., 2019) | 94.53 | 63.30 | 40.23 | 18.11 | 68.92 | 56.21 |
| FGSM | BANANAS (White et al., 2021a) | 93.52 | 66.35 | 45.59 | 20.72 | 68.01 | 54.88 |
| | Local Search (White et al., 2021b) | 93.86 | 69.10 | 48.27 | 23.18 | 69.47 | 56.57 |
| | Random Search (Li & Talwalkar, 2019) | 93.57 | 67.25 | 46.15 | 20.93 | 68.44 | 55.10 |
| | Regularized Evolution (Real et al., 2019) | 93.77 | 68.82 | 47.99 | 22.59 | 69.20 | 56.11 |

## 4.2 NAS ON ROBUSTNESS

In this section, we perform different state-of-the-art NAS algorithms on the clean accuracy and the FGSM ($\epsilon = 1$) robust accuracy in the NAS-Bench-201 (Dong & Yang, 2020) search space, and evaluate the best found architectures on all provided introduced adversarial attacks. We apply random search (Li & Talwalkar, 2019), local search (White et al., 2021b), regularized evolution (Real et al., 2019) and BANANAS (White et al., 2021a) with a maximal query amount of 300. The results are shown in Table 1. Although clean accuracy is reduced, the overall robustness to all adversarial attacks improves when the search is performed on FGSM ($\epsilon = 1.0$) accuracy. Local Search achieves the best performance, which indicates that localized changes to an architecture design seem to be able to improve network robustness.

## 4.3 ANALYZING THE EFFECT OF ARCHITECTURE DESIGN ON ROBUSTNESS

In Figure 7, we show the top-20 performing architectures (color-coded, one operation for each edge) with exactly 2 times $3 \times 3$ convolutions and no $1 \times 1$ convolutions (hence, the same parameter count), according to the mean adversarial accuracy over all attacks as described in subsection 3.2 on

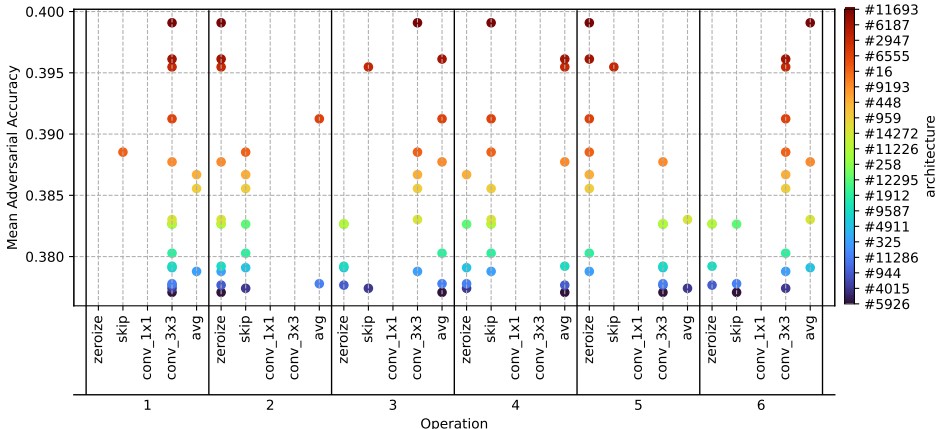

Figure 7: Top-20 architectures (out of $408$) with exactly 2 times $3 \times 3$ convolutions and no $1 \times 1$ convolutions according to mean adversarial accuracy on CIFAR-10. The operation number (1-6) corresponds to the edge in the cell, see Figure 1 for cell connectivity and operations. Stacking convolutions seems to be an important part of robust architectural design.

CIFAR-10. It is interesting to see that there are no convolutions on edges 2 and 4, and additionally no dropping (operation zeroize) or skipping (operation skip-connect) of edge 1. In the case of edge 4, it seems that a single convolutional layer connecting input and output of the cell increases sensitivity of the network. Hence, most of the top-20 robust architectures stack convolutions (via edge 1, followed by either edge 3 or 5), from which we hypothesize that stacking convolution operations might improve robustness when designing architectures. At the same time, skipping input to output via edge 4 seems not to affect robustness negatively, as long as the input feature map is combined with stacked convolutions. Further analyses can be found in Appendix B. We find that optimizing architecture design can have a substantial impact on the robustness of a network. In this setting, where networks have *the same parameter count*, we can see a large range of mean adversarial accuracies $[0.21, 0.4]$ showing the potential of doubling the robustness of a network by carefully crafting its topology. Important to note here is that this is a first observation, which can be made by using our provided dataset. This observation functions as a motivation for how this dataset can be used to analyze robustness in combination with architecture design.

## 5 CONCLUSION

We introduce a dataset for neural architecture design and robustness to provide the research community with more resources for analyzing what constitutes robust networks. We have evaluated *all* $6\,466$ unique architectures from the commonly used NAS-Bench-201 benchmark against several adversarial attacks and image dataset corruptions. With this full evaluation at hand, we presented three use cases for this dataset: First, the correlation between the robustness of the architectures and two differentiable architecture measurements. We showed that these measurements are a good first approach for the architecture's robustness, but have to be taken with caution when the perturbation increases. Second, neural architecture search directly on the robust accuracies, which indeed finds more robust architectures for different adversarial attacks. And last, an initial analysis of architectural design, where we showed that it is possible to improve robustness of networks with the same number of parameters by carefully designing their topology.

**Limitations and Broader Impact** We show that carefully crafting the architectural design can lead to substantial impact on the architecture's robustness. This paper aims to promote and streamline research on how the *architecture design* affects model robustness. It does so by evaluating pretrained architectures and is thus complementary to any work focusing on the analysis of different, potentially adversarial, training protocols.

## REPRODUCIBILITY

In order to ensure reproducibility, we build our dataset on architectures from a common NAS-Benchmark, which is described in the main paper in subsection 3.1. In addition, all hyperparameters for reproducing the robustness results are given in subsection 3.2. Lastly, a complete description of the dataset itself is provided in the appendix, Appendix A.

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

# A DATASET

## A.1 NAS-BENCH-201

We base our evaluations on the NAS-Bench-201 (Dong & Yang, 2020) search space. It is a cell-based architecture search space. Each cell has in total $4$ nodes and $6$ edges. The nodes in this search space correspond to the architecture's feature maps and the edges represent the architectures operation, which are chosen from the operation set $\mathcal{O} = \{1 \times 1 \,\mathrm{conv.} \,, 3 \times 3 \,\mathrm{conv.} \,, 3 \times 3 \,\mathrm{avg.\ pooling} \,, \mathrm{skip} \,, \mathrm{zero}\}$ (see Figure 1). This search space contains in total $5^6 = 15\,625$ architectures, from which only $6\,466$ are unique, since the operations skip and zero can cause isomorphic cells (see Figure 8), where the latter operation zero stands for dropping the edge. Each architecture is trained on three different image datasets for 200 epochs: CIFAR-10 (Krizhevsky, 2009), CIFAR-100 (Krizhevsky, 2009) and ImageNet16-120 (Chrabaszcz et al., 2017). For our evaluations, we consider all unique architectures in the search space and test splits of the corresponding datasets. Hence, we evaluate $3 \cdot 6\,466 = 19\,398$ pretrained networks in total.

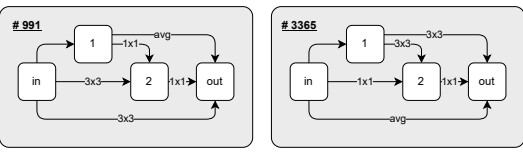

Figure 8: Example of two isomorphic graphs in NAS-Bench-201. Due to the skip connection from node in to node 1, both computational graphs are equivalent, but their identification in the search space is different. For this dataset, we evaluated all non-isomorphic graphs (#991 was evaluated and #3365 was not).

## A.2 DATASET GATHERING

We collect evaluations for our dataset for different corruptions and adversarial attacks (as discussed in subsection 3.2 and subsection 3.3) following algorithm 1. This process is also depicted in Figure 9. Hyperparameter settings for adversarial attacks are listed in Table 2. Due to the heavy load of running all these evaluations, they are performed on several clusters. These clusters are comprised of either (i) compute nodes with Nvidia A100 GPUs, 512 GB RAM, and Intel Xeon IceLake-SP processors, (ii) compute nodes with NVIDIA Quadro RTX 8000 GPUs, 1024 GB RAM, and AMD EPYC 7502P processors, (iii) NVIDIA Tesla A100 GPUs, 2048 GB RAM, Intel Xeon Platinum 8360Y processors, and (iv) NVIDIA Tesla A40 GPUs, 2048 GB RAM, Intel Xeon Platinum 8360Y processors.

Table 2: Hyperparameter settings of adversarial attacks evaluated.

| Attack | Hyperparameters |
|---|---|
| FGSM | $\epsilon \in \{.1, .5., 1, 2, 3, 4, 5, 6, 7, 8, 255\}/255$ |
| PGD | $\epsilon \in \{.1, .5., 1, 2, 3, 4, 8, 255\}/255$ 
 $\alpha = 0.01/0.3$ 
 40 attack iterations |
| APGD | $\epsilon \in \{.1, .5., 1, 2, 3, 4, 8, 255\}/255$ 
 100 attack iterations |
| Square | $\epsilon \in \{.1, .5., 1, 2, 3, 4, 8, 255\}/255$ 
 5 000 search iterations |

---

**Algorithm 1:** Robustness Dataset Gathering

---

**Input:** (i) Architecture space $A$ (NAS-Bench-201).
**Input:** (ii) Test datasets $D$ (CIFAR-10, CIFAR-100, ImageNet16-120).
**Input:** (iii) Set of attacks and/or corruptions $C$.
**Input:** (iv) Robustness Dataset $R$.

**1** **for** $a \in A$ **do**
  ▷ Load pretrained weights for $a$.
**2**   $a$.load_weights($d$)
**3**   **for** $d \in D$ **do**
**4**     **for** $c(\cdot,\cdot) \in C$ **do**
        ▷ Corrupt dataset $d$.
**5**       $d_c \leftarrow c(a,d)$
        ▷ Evaluate architecture $a$ with $d_c$.
**6**       $\text{Accuracy}, \text{Confidence}, \text{ConfusionMatrix} \leftarrow eval(a, d_c)$
        ▷ Extend robustness dataset with evaluations.
**7**       $R[d][c][\text{"accuracy"}][a] \leftarrow \text{Accuracy}$
**8**       $R[d][c][\text{"confidence"}][a] \leftarrow \text{Confidence}$
**9**       $R[d][c][\text{"cm"}][a] \leftarrow \text{ConfusionMatrix}$
**10**     **end**
**11**   **end**
**12** **end**

---

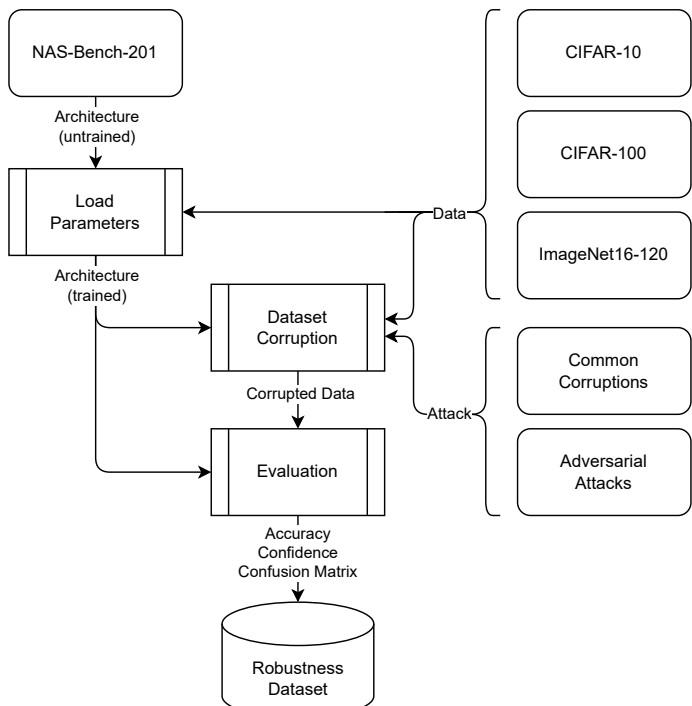

Figure 9: Diagram showing the gathering process for our robustness dataset. (i) An non-isomorphic architecture contained in NAS-Bench-201 is created and its parameters are loaded from a provided checkpoint, dependent on the dataset evaluated. (ii) Given the evaluation dataset, an attack or corruption, and the trained network, the evaluation dataset is corrupted and (iii) the resulting corrupted data is used to evaluate the network. (iv) The evaluation results are stored in our robustness dataset.

### A.3 Dataset Structure, Distribution, and License

Files are provided in `json` format to ensure platform-independence and to reduce the dependency on external libraries (e.g. Python has built-in `json`-support).

We will publish code that accompanies our dataset on GitHub. The dataset itself will be linked from GitHub and is hosted on an institutional cloud service. This ensures longtime availability and the possibility to version the dataset. Dataset and code will be published at the notification date under GNU GPLv3.

### A.4 Structure

The dataset consists of 3 folders, one for each dataset evaluated (`cifar10`, `cifar100`, `ImageNet16-120`). Each folder contains one `json` file for each combination of key and measurement. Keys refer to the sort of attack or corruption used (Table 3 lists all keys). Measurements refer to the collected evaluation type (`accuracy`, `confidence`, `cm`). Clean and adversarial evaluations are performed on all datasets, while common corruptions are evaluated on `cifar10` and `cifar100`. Additionally, the dataset contains one metadata file (`meta.json`).

Table 3: Keys for attacks and corruptions evaluated.

| Clean | Adversarial | Common Corruptions |
|-------|-------------|--------------------|
| clean | aa_apgd-ce  | brightness         |
|       | aa_square   | contrast           |
|       | fgsm        | defocus_blur       |
|       | pgd         | elastic_transform  |
|       |             | fog                |
|       |             | frost              |
|       |             | gaussian_noise     |
|       |             | glass_blur         |
|       |             | impulse_noise      |
|       |             | jpeg_compression   |
|       |             | motion_blur        |
|       |             | pixelate           |
|       |             | shot_noise         |
|       |             | snow               |
|       |             | zoom_blur          |

**Metadata**  The `meta.json` file contains information about each architecture in NAS-Bench-201. This includes, for each architecture identifier, the corresponding string defining the network design (as per (Dong & Yang, 2020)) as well as the identifier of the corresponding non-isomorphic architecture from (Dong & Yang, 2020) that we evaluated. The file also contains all $\epsilon$ values that we evaluated for each adversarial attack. An excerpt of this file is shown in Figure 10.

**Files**  All files are named according to `"{key}_{measurement}.json"`. Hence, the path to all clean accuracies on `cifar10` is `"./cifar10/clean_accuracy.json"`. An excerpt of this file is shown in Figure 11. Each file contains nested dictionaries stating the dataset, evaluation key and measurement type. For evaluations with multiple measurements, e.g. in the case of adversarial attacks for multiple $\epsilon$ values, the results are concatenated into a list. Files and their possible contents are described in Table 4.

```
{
  "ids": {
    ...,
    "21": {
      "nb201-string": "|nor_conv_1x1~0|+|none~0|none~1|+|nor_conv_1x1~0|nor_conv_3x3~1|none~2|",
      "isomorph": "21"
    },
    ...,
    "1832": {
      "nb201-string": "|nor_conv_1x1~0|+|nor_conv_1x1~0|none~1|+|nor_conv_1x1~0|skip_connect~1|none~2|",
      "isomorph": "309"
    },
    ...
  },
  "epsilons": {
    "aa_apgd-ce": [0.1, 0.5, 1.0, 2.0, 3.0, 4.0, 8.0],
    "aa_square": [0.1, 0.5, 1.0, 2.0, 3.0, 4.0, 8.0],
    "fgsm": [0.1, 0.5, 1.0, 2.0, 3.0, 4.0, 5.0, 6.0, 7.0, 8.0, 255.0],
    "pgd": [0.1, 0.5, 1.0, 2.0, 3.0, 4.0, 8.0]
  }
}
```

Figure 10: Excerpt of `meta.json` showing meta information of architectures #21 and #1832, as well as $\epsilon$ values for each attack. Architecture #21 is non-isomorphic and points to itself, while architecture #1832 is an isomorphic instance of #309.

```
{                                    {
  "cifar10": {                         "cifar10": {
    "clean": {                           "pgd": {
      "accuracy": {                        "accuracy": {
        "0": 0.856,                          "0": [0.812, 0.582, 0.295, 0.034, 0.002, 0.0, 0.0],
        ...                                  ...
      }                                    }
    }                                    }
  }                                    }
}                                    }
```

Figure 11: Excerpt of **(left)** `clean_accuracy.json` and **(right)** `pgd_accuracy.json` for dataset `cifar10` for the architecture #0. Numbers are rounded to improve readability.

Table 4: Files and their possible content.

| File | Description |
|---|---|
| clean_accuracy | one accuracy value for each evaluated network |
| clean_confidence | one confidence matrix for each evaluated network and collection scheme |
| clean_cm | one confusion matrix for each evaluated network |
| {attack}_accuracy | list of accuracies, where each element corresponds to the respective $\epsilon$ value |
| {attack}_confidence | list of confidence matrices, where each element corresponds to the respective $\epsilon$ value |
| {attack}_cm | list of confusion matrices, where each element corresponds to the respective $\epsilon$ value |
| {corruption}_accuracy | list of accuracies, where each element corresponds to the respective corruption severity |
| {corruption}_confidence | list of confidence matrices, where each element corresponds to the respective corruption severity |
| {corruption}_cm | list of confusion matrices, where each element corresponds to the respective corruption severity |

We showed some analysis and possible use-cases on accuracies in the main paper. In the following, we elaborate on and show `confidence` and confusion matrix (`cm`) measurements.

## A.5 CONFIDENCE

We collect the mean confidence after softmax for each network over the whole (attacked) test dataset evaluated. We used 3 schemes to collect confidences (see Figure 13). First, confidences for each class are given by true labels (called `label`). In case of `cifar10`, this results in a $10 \times 10$ confidence matrix, for `cifar100` a $100 \times 100$ confidence matrix, and `ImageNet16-120` a $120 \times 120$ confidence matrix. Second, confidences for each class are given by the class predicted by the network (called `argmax`). This again results in matrices of sizes as mentioned. Third, confidences for correctly classified images as well as confidences for incorrectly classified images (called `prediction`). For all image datasets, this results in a vector with 2 dimensions. Each result is saved as a list (or list of list), see Figure 12.

Figure 14 shows a progression of `label` confidence values for class label 0 on `cifar10` from `clean` to `fgsm` with increasing values of $\epsilon$. Figure 15 shows how `prediction` confidences of correctly and incorrectly classified images correlate with increasing values of $\epsilon$ when attacked with `fgsm`.

```
{
  "cifar10": {
    "clean": {
      "confidence": {
        "0": {
          "label": [[...]],
          "argmax": [[...]],
          "prediction": [...]
        }
      }
    }
  }
}
```

Figure 12: Excerpt of `clean_confidence.json` for `cifar10`. Numbers are not shown to improve readability.

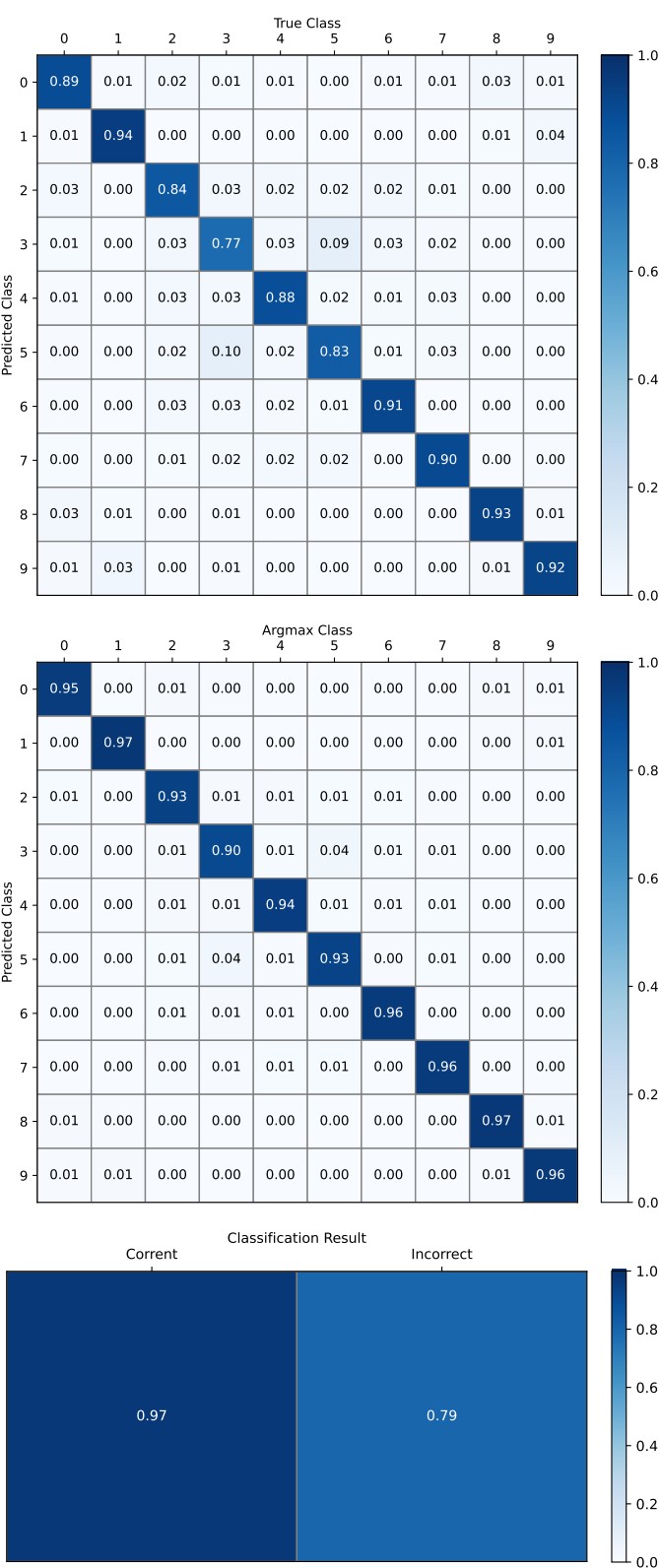

Figure 13: Mean confidence scores on clean CIFAR-10 images for all non-isomorphic networks in NAS-Bench-201. (**top**: `label`) For each true class label. (**middle**: `argmax`) For each predicted class label. (**bottom**: `prediction`) For correct and incorrect classifications.

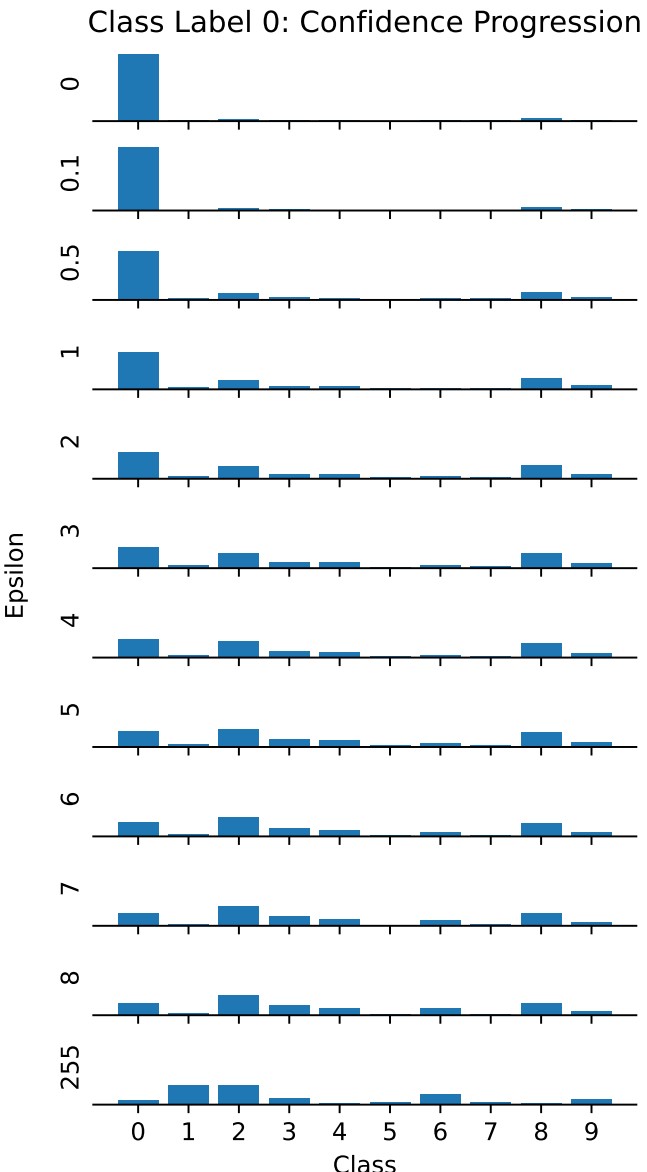

Figure 14: Mean `label` confidence scores on FGSM-attacked CIFAR-10 images for different $\epsilon$ for all non-isomorphic networks in NAS-Bench-201. Only confidence scores for class label 0 are shown. Networks lose prediction confidence for the true label when $\epsilon$ increases.

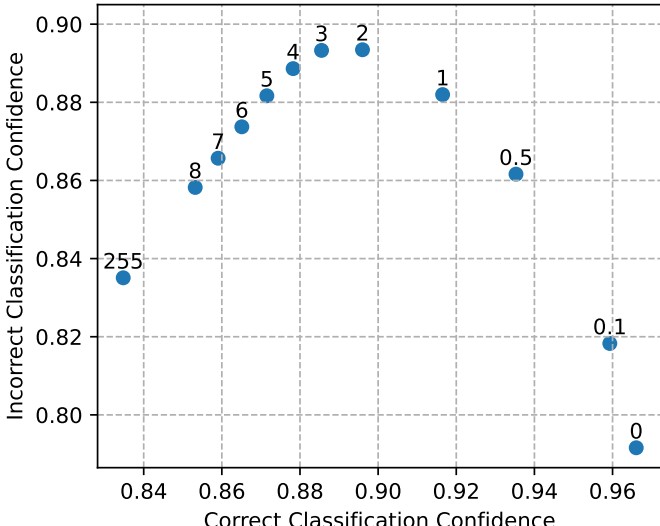

Figure 15: Mean `prediction` confidence scores on FGSM-attacked CIFAR-10 images for different $\epsilon$ (on top of points) for all non-isomorphic networks in NAS-Bench-201. Networks become less confident in their prediction if their prediction is correct when $\epsilon$ increases. Networks become more confident in their prediction if their prediction is incorrect, however, only up to a certain $\epsilon$ value. When $\epsilon$ further increases, confidence drops again.

## A.6 CONFUSION MATRIX

For each evaluated network, we collect the confusion matrix (key: `cm`) for the corresponding (attacked) test dataset. The result is a $10 \times 10$ matrix in case of `cifar10`, a $100 \times 100$ matrix in case of `cifar100`, and a $120 \times 120$ matrix in case of `ImageNet16-120`. See Figure 16 for an example, where we summed up confusion matrices for all networks on `cifar10`.

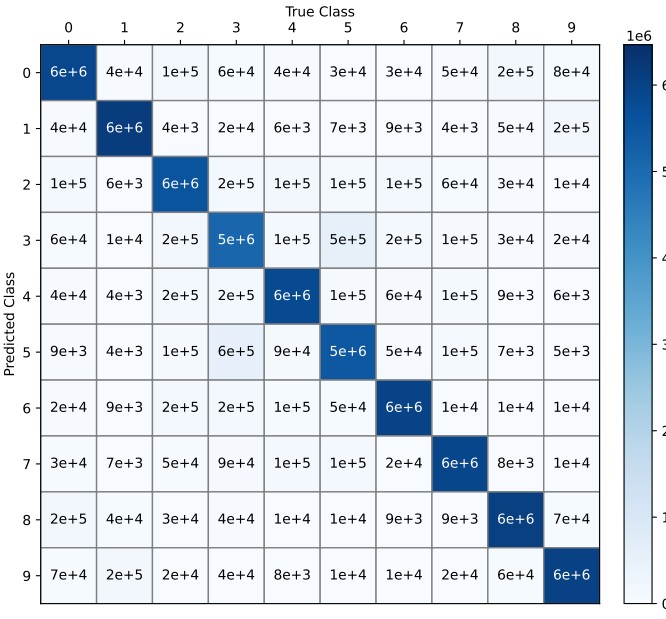

Figure 16: Aggregated confusion matrices on clean CIFAR-10 images for all non-isomorphic networks in NAS-Bench-201.

## A.7  CORRELATIONS BETWEEN IMAGE DATASETS

In Figure 17 we show the correlation between all clean and adversarial accuracies over all datasets collected. This plot shows a positive correlation between the image datasets for the one-step FGSM attack, whereas for all other multi-step attacks, the correlation becomes close to zero or even negative.

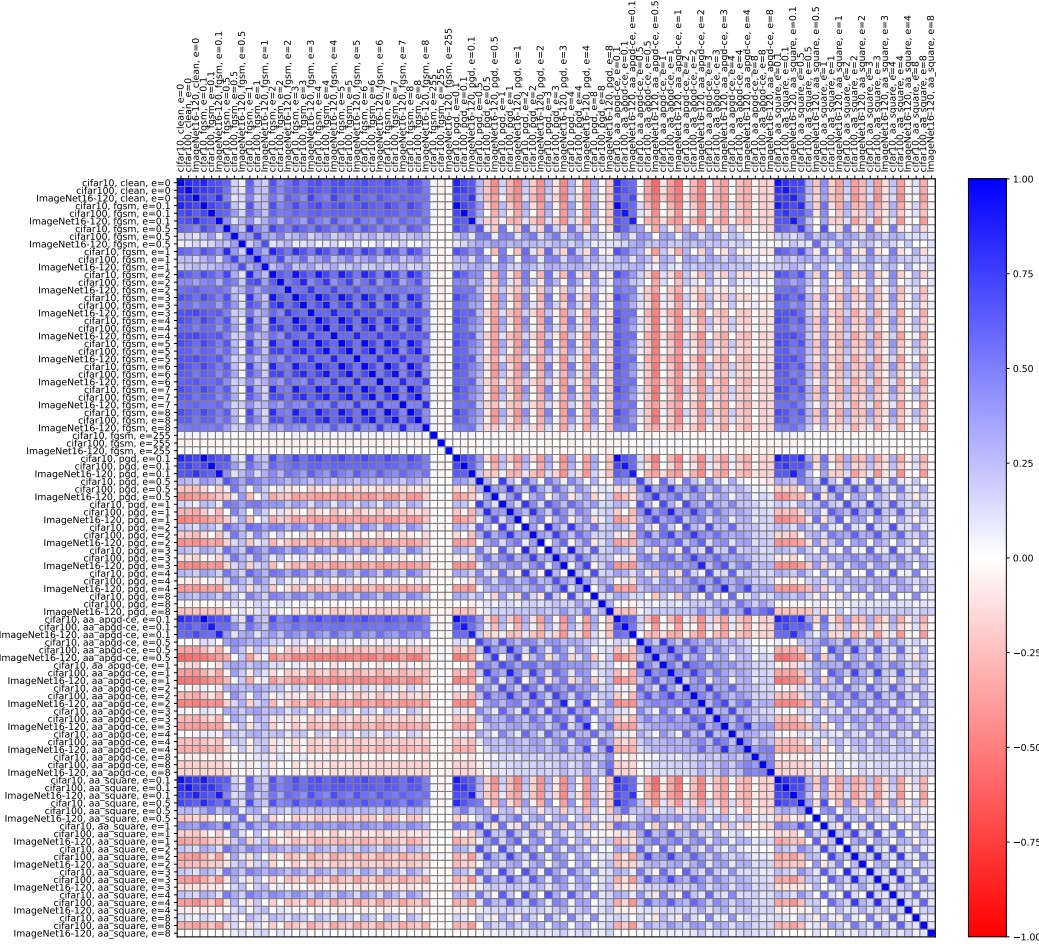

Figure 17:  Kendall rank correlation coefficient between all clean and adversarial accuracies that are evaluated in our dataset.

## A.8 EXAMPLE IMAGE OF CORRUPTIONS IN CIFAR-10-C

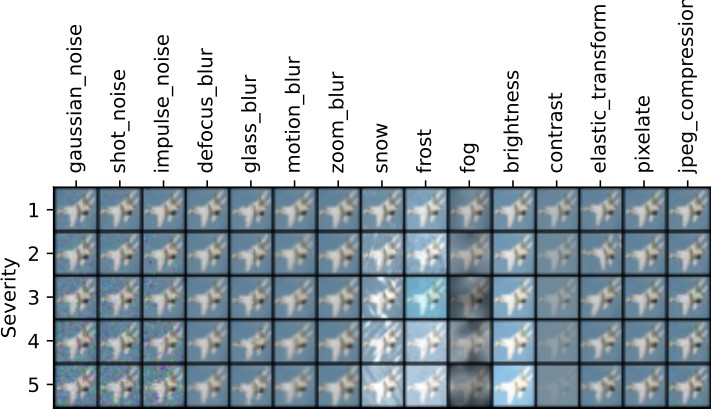

Figure 18: An example image of CIFAR-10-C (Hendrycks & Dietterich, 2019) with different corruption types at different severity levels. CIFAR-100-C (Hendrycks & Dietterich, 2019) consists of images with the same corruption types and severity levels.

## A.9 MAIN PAPER FIGURES FOR OTHER IMAGE DATASETS

### A.9.1 CIFAR-100 ADVERSARIAL ATTACK ACCURACIES (FIGURE 2)

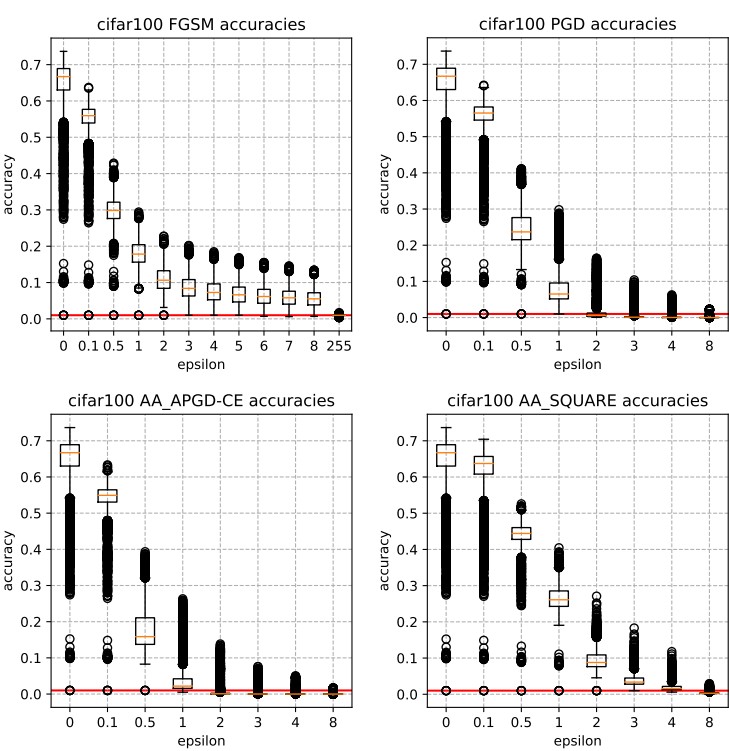

Figure 19: Accuracy boxplots over all unique architectures in NAS-Bench-201 for different adversarial attacks (FGSM (Goodfellow et al., 2015), PGD (Kurakin et al., 2017), APGD (Croce & Hein, 2020), Square (Andriushchenko et al., 2020)) and perturbation magnitude values $\epsilon$, evaluated on CIFAR-100. Red line corresponds to guessing.

### A.9.2 IMAGENET16-120 ADVERSARIAL ATTACK ACCURACIES (FIGURE 2)

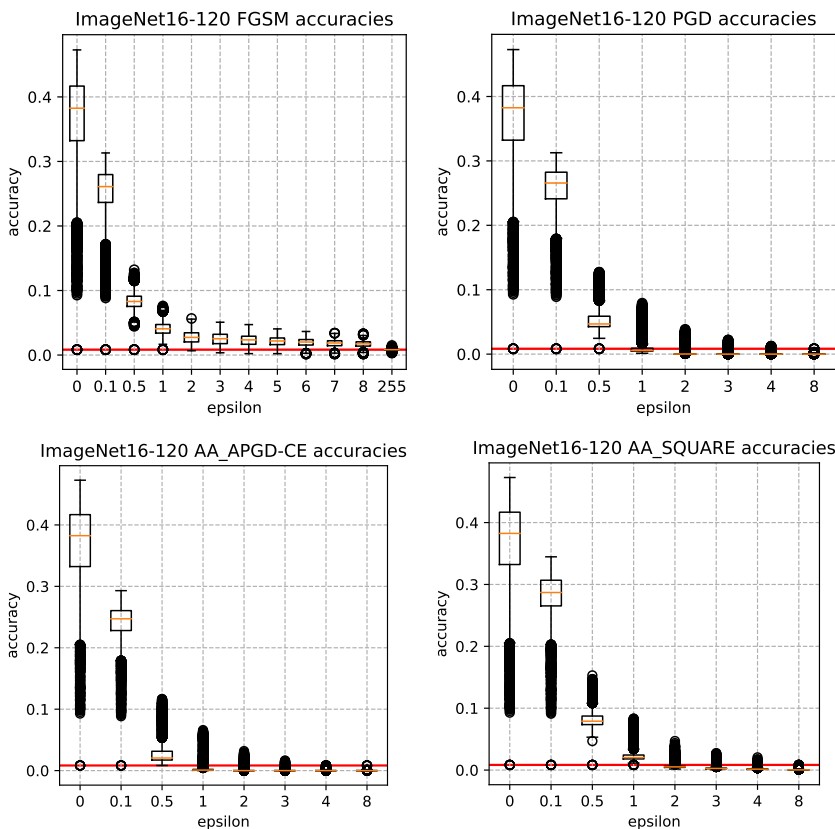

Figure 20: Accuracy boxplots over all unique architectures in NAS-Bench-201 for different adversarial attacks (FGSM (Goodfellow et al., 2015), PGD (Kurakin et al., 2017), APGD (Croce & Hein, 2020), Square (Andriushchenko et al., 2020)) and perturbation magnitude values $\epsilon$, evaluated on ImageNet16-120. Red line corresponds to guessing.

## A.9.3 CIFAR-10-C COMMON CORRUPTION ACCURACIES (FIGURE 4)

Figure 21: Accuracy boxplots over all unique architectures in NAS-Bench-201 for different corruption types at different severity levels, evaluated on CIFAR-10-C. Red line corresponds to guessing.

A.9.4 CIFAR-100-C COMMON CORRUPTION ACCURACIES (FIGURE 4)

Figure 22: Accuracy boxplots over all unique architectures in NAS-Bench-201 for different corruption types at different severity levels, evaluated on CIFAR-100-C. Red line corresponds to guessing.

### A.9.5 CIFAR-100 ADVERSARIAL ATTACK CORRELATIONS (FIGURE 3)

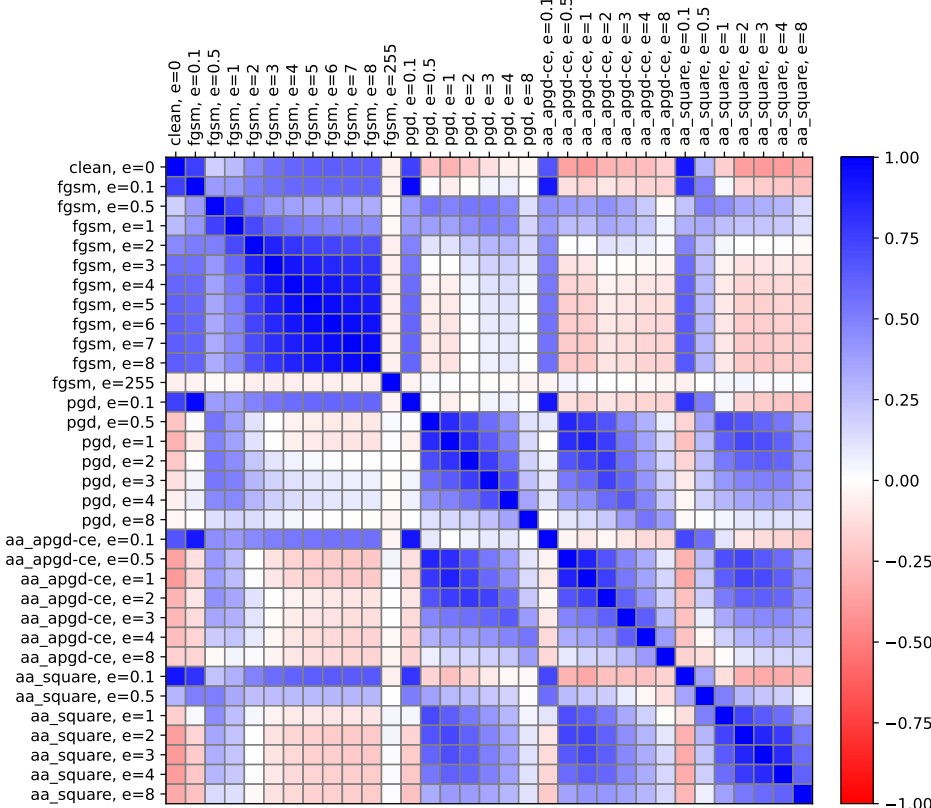

Figure 23: Kendall rank correlation coefficient between clean accuracies and robust accuracies on different attacks and magnitude values $\epsilon$ on CIFAR-100 for all unique architectures in NAS-Bench-201.

### A.9.6 IMAGENET16-120 ADVERSARIAL ATTACK CORRELATIONS (FIGURE 3)

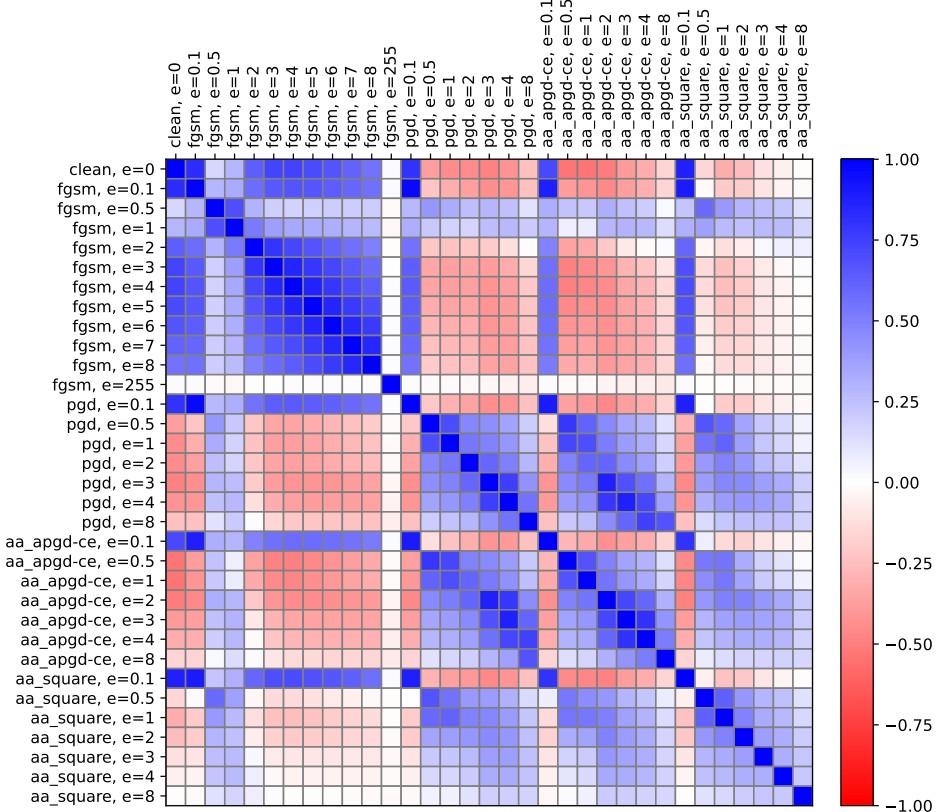

Figure 24: Kendall rank correlation coefficient between clean accuracies and robust accuracies on different attacks and magnitude values $\epsilon$ on ImageNet16-120 for all unique architectures in NAS-Bench-201.

### A.9.7 CIFAR-100-C COMMON CORRUPTION CORRELATIONS (FIGURE 5)

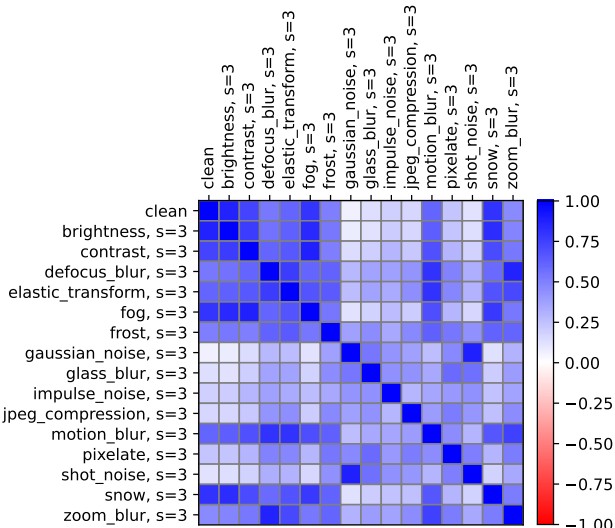

Figure 25: Kendall rank correlation coefficient between clean accuracies and accuracies on different corruptions at severity level 4 on CIFAR-100-C for all unique architectures in NAS-Bench-201.

# B ANALYSIS

In this section, we first depict the best architectures in NAS-Bench-201 (Dong & Yang, 2020) in subsection B.1, then show the effect of parameter count on robustness and the magnitude of potential gains in robustness in a limited parameter count setting in subsection B.2, and lastly show the effect of single changes to the best performing architecture according to clean accuracy in subsection B.3.

## B.1 BEST ARCHITECTURES

Figure 26 visualizes the best architectures in the NAS-Bench-201 (Dong & Yang, 2020) search space in terms of clean accuracy, mean adversarial accuracy, and mean common corruption accuracy on CIFAR-10 and their respective edit distances. The edit distance is defined by the number of changes, either node or edge, to change the graph to the target graph. In the case of NAS-Bench-201 architectures, an edit distance of 1 means that exactly one operation differs between two architectures. So in order to modify the best performing architecture in terms of clean accuracy (#13714) into the best performing architecture according to mean corruption accuracy (#3456), we need to exchange two (out of six) operations: (i) exchange operation 2 from $3 \times 3$ convolution to zero and (ii) exchange operation 5 from $1 \times 1$ convolution to $3 \times 3$ convolution.

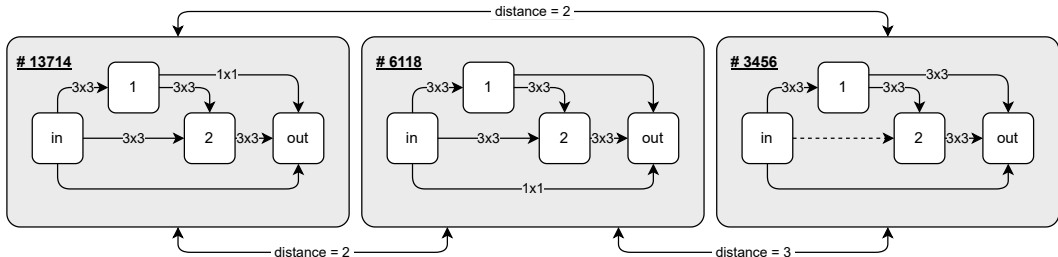

Figure 26: Best architectures in NAS-Bench-201 according to (**left**) clean accuracy, (**middle**) mean adversarial accuracy (over all attacks and $\epsilon$ values as described in subsection 3.2), and (**right**) mean common corruption accuracy (over all corruptions and severities) on CIFAR-10. See Figure 1 for cell connectivity and operations.

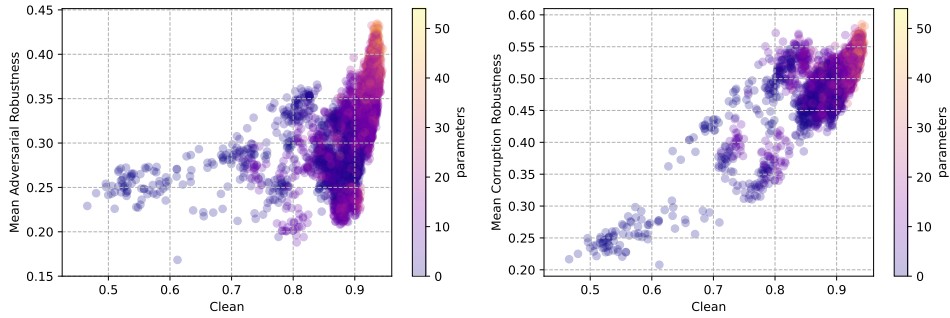

Figure 27: (**left**) Mean adversarial robustness accuracies and (**right**) mean corruption robustness accuracies vs. clean accuracies on CIFAR-10 for all unique architectures in NAS-Bench-201. Scatter points are colored based on the number of kernel parameters of a single cell (1 for each $1 \times 1$ convolution, 9 for each $3 \times 3$ convolution).

## B.2 CELL KERNEL PARAMETER COUNT

Figure 27 displays the mean adversarial robustness accuracies (left) and the mean corruption robustness accuracies (right) against the clean accuracy, color-coded by the number of cell kernel parameters. We count 1 for each $1 \times 1$ convolution and 9 for each $3 \times 3$ convolution contained in

the cell, hence, their number ranges in $[0, 54]$. Since these are multipliers for the parameter count of the whole network, we coin these *cell kernel parameters*. Overall, we can see that the cell kernel parameter count matters in terms of robustness, hence, that networks with large parameter counts are more robust in general. We can also see that the number of cell kernel parameters are more essential for robustness against common corruptions, where the correlation between clean and corruption accuracy is more linear. Also in terms of adversarial robustness, there seems to be a large magnitude of possible improvements that can be gained by optimizing architecture design.

**Limited Cell Parameter Count**    To further investigate the magnitude of possible improvements via architectural design optimization, we look into the scenario of limited cell parameter count.

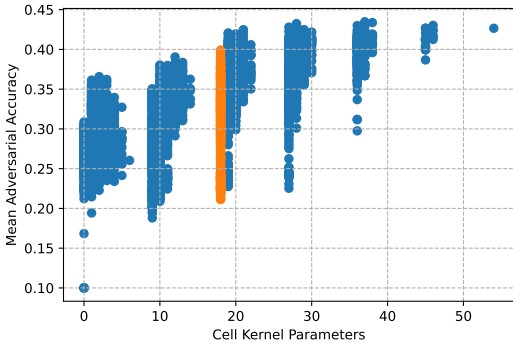

Figure 28: Mean robust accuracy over all attacks as described in subsection 3.2 on CIFAR-10 by kernel parameters $\in [0, 54]$ for all unique architectures in NAS-Bench-201. Orange scatter points depict all architectures with kernel parameter count 18, hence, architectures with exactly 2 times $3 \times 3$ convolutions. Although having exactly the same parameter count, the mean adversarial robustness of these networks ranges in $[0.21, 0.40]$.

In Figure 28, we depict all unique architectures in NAS-Bench-201 by their mean adversarial robustness and cell kernel parameter count. Networks with parameter count 18 (408 instances in total) are highlighted in orange. As we can see, there is a large range of mean adversarial accuracies $[0.21, 0.4]$ for the parameter count 18 showing the potential of doubling the robustness of a network with *the same parameter count* by carefully crafting its topology. In Figure 29 we show the top-20 performing architectures (color-coded, one operation for each edge) in the mentioned scenario of a parameter count of 18, according to (**top**) mean adversarial and (**bottom**) mean corruption accuracy. It is interesting to see that in both cases, there are (almost) no convolutions on edges 2 and 4, and additionally no dropping or skipping of edge 1. In the case of edge 4, it seems that a single convolution layer connecting input and output of the cell increases sensitivity of the network. Hence, most of the top-20 robust architectures stack convolutions (via edge 1, followed by either edge 3 or 5), from which we hypothesize that stacking convolutions operations might improve robustness when designing architectures. At the same time, skipping input to output via edge 4 seems not to affect robustness negatively, as long as the input feature map is combined with stacked convolutions. Important to note here is that this is a first observation, which can be made by using our provided dataset. This observation functions as a motivation for how this dataset can be used to analyze robustness in combination with architecture design.

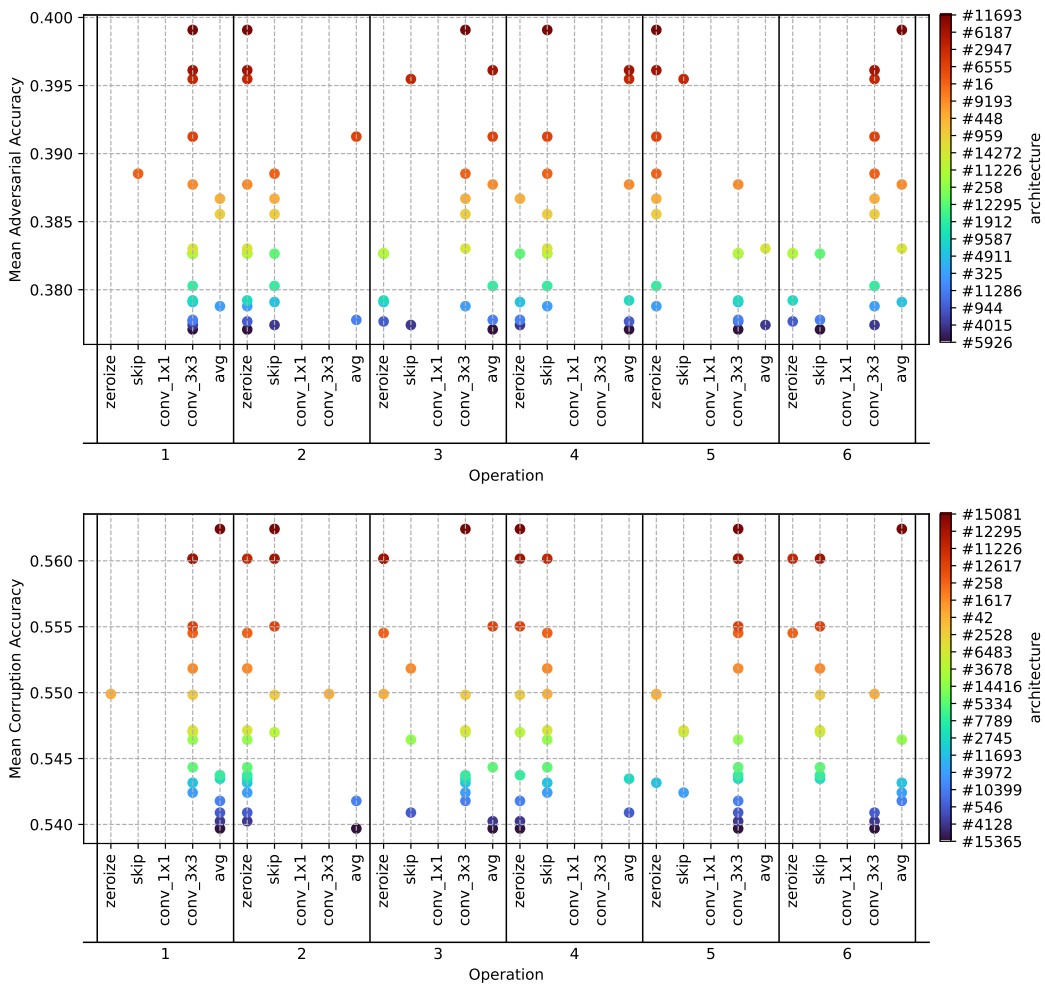

Figure 29: Top-20 architectures with cell kernel parameter count 18 (hence, architectures with exactly 2 times $3 \times 3$ convolutions) according to (**top**) mean adversarial accuracy and (**bottom**) mean corruption accuracy on CIFAR-10. See Figure 1 for cell connectivity and operations (1-6).

### B.3 GAINS AND LOSSES BY SINGLE CHANGES

The fact that our dataset contains evaluations for all unique architectures in NAS-Bench-201 enables us to analyze the effect of small architectural changes. In Figure 30, we depict again all unique architectures by their clean and robust accuracies on CIFAR-10 (Krizhevsky, 2009). The red data point in both plots shows the best performing architecture in terms of clean accuracy (#13714, see Figure 26), while the orange points are its neighboring architectures with edit distance 1. The operation changed for each point is shown in the legend. As we can see in the case of adversarial attacks, we can trade-off more robust accuracy for less clean accuracy by changing only one operation. While some changes seem obvious (adding more parameters as with 13 and 14), it is interesting to see that exchanging the $3 \times 3$ convolution on edge 3 with average pooling (and hence, reducing the amount of parameters) also improves adversarial robustness. In terms of robustness towards common corruptions, each architectural change leads to worse clean and robust accuracy in this case. Changing more than one operation is necessary to improve common corruption accuracy of this network (as we have seen in Figure 26).

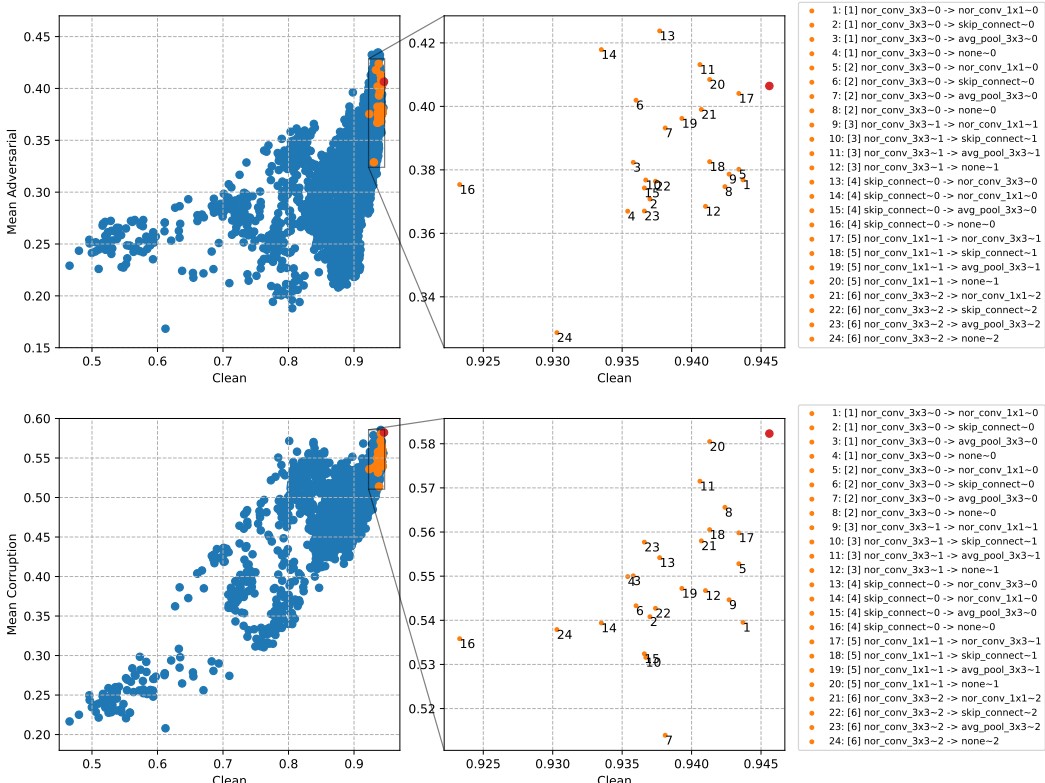

Figure 30: (**top**) Scatter plot clean accuracy vs. mean adversarial accuracy (over all attacks and $\epsilon$ values as described in subsection 3.2) on CIFAR-10. (**bottom**) Scatter plot clean accuracy vs. mean common corruption accuracy (over all corruptions and severities) on CIFAR-10. The red data point shows the best performing architecture according to clean accuracy on CIFAR-10. The orange data points are neighboring architectures, where exactly one operation differs. The change of operation is depicted in the legend. The number in brackets refers to the edge where the operation was changed. See Figure 1 for cell connectivity and operations (1-6).

