# OpenReview forum: "Neural Architecture Design and Robustness: A Dataset"
_ICLR.cc/2023/Conference — ICLR 2023 poster_

### Official Review · Reviewer_HhBB · 2022-10-13

**Confidence:** 4
**Correctness:** 3
**Technical Novelty And Significance:** 2
**Empirical Novelty And Significance:** 3
**Recommendation:** 6

**Clarity, Quality, Novelty And Reproducibility:**

Clarity: 7/10

Quality: 6/10

Novelty: 5/10

Reproducibility: 8/10

**Strength And Weaknesses:**

Strengths:
1. The tackled problem is relevant to the ICLR community.
2. Several experiments have been conducted, and comprehensive results are provided.

Weaknesses:
1. From the technical perspective, no new concepts have been introduced.
2. It would be useful to describe the main methodology through algorithms and schemes.

**Summary Of The Paper:**

This work extends the NAS-Bench-201 dataset to evaluate the neural architecture design and robustness. Moreover, an analysis of how the architectural design affects robustness is conducted.

**Summary Of The Review:**

A borderline paper with few technical novel contents but solid evaluation and results.

---

> ### Author Response · Authors · 2022-11-12
> **Reply**
>
> We appreciate the reviewer’s recognition of our work and the suggestions made.
>
> **1. From the technical perspective, no new concepts have been introduced.**
>
> The motivation for this dataset is to provide the possibility to investigate the effects of architectural changes, which is an established paradigm in neural architecture search. It is suited to apply both black-box optimization based approaches as well as gradient-based architecture optimization schemes, and has been strongly argued for for example in [1]. We facilitate this by providing robustness evaluations for the complete search space of NAS-Bench-201 for our research community. There is no work (that we know of) that evaluated a **whole NAS search space** for robustness so far. By providing this dataset, we enable researchers to go beyond the plain evaluation and optimization with respect to accuracy and facilitate the analysis of the impact of architecture modifications on robustness, which we consider one of the crucial current challenges for sustainable deep learning.
>
> [1]  Dong, et al., NATS-Bench: Benchmarking NAS Algorithms for Architecture Topology and Size, https://arxiv.org/abs/2009.00437
>
> **2. It would be useful to describe the main methodology through algorithms and schemes.**
>
> We added an algorithm as well as a diagram explaining how we collected the data for our dataset in Appendix A.2. Is that what you meant?
>
> We hope we addressed all remaining concerns and look forward to further questions and discussion.

---

> > ### Comment · Reviewer_HhBB · 2022-11-21
> > **Response to Authors**
> >
> > The efforts made by the authors in answering the reviewers' comments are appreciated. The score is confirmed.

---

> ### Author Response · Authors · 2022-11-16
> **Follow Up**
>
> Dear Reviewer,
>
> Since the window of making adjustments to our manuscript closes on Friday, please let us know if our changes resolved your remaining concerns and if you are willing to raise your score.
> Is there anything else we should consider?
>
> Thank you,
>
> Authors of Paper #2665

---

### Official Review · Reviewer_VzZD · 2022-10-24

**Confidence:** 2
**Correctness:** 4
**Technical Novelty And Significance:** 3
**Empirical Novelty And Significance:** 3
**Recommendation:** 8

**Clarity, Quality, Novelty And Reproducibility:**

Clearly written, novelty is medium high, and artifacts are provided for reproducibility.

**Strength And Weaknesses:**

-= S1 =- The paper extends the NAS benchmark by adding the important dimension of robustness to it. This empirical study is well-motivated, the experiments are presented in a nice and descriptive manner, and this dataset can help other researchers and designers to design more robust models.

-= S2 =- The paper is very well-written and provide a nice background on various adversarial attacks that can be leveled against a model. Additionally, the results are presented in an easy to understand manner.

-= S3 =- The three use cases presented for this dataset are also well-motivated and evaluated.

-= S4 =- Toward the end of the paper, the authors provide a set of empirical analyses on what aspects of the network are highly influential on robustness. These observations can help future researchers and designers of network.

-= W1 =- The paper can benefit from providing more descriptive captions to the figures. It will be nice if the figures can be read in a standalone manner i.e., they explain what the results are.

-= W2 =- The paper can benefit from discussing the results a little bit more. For instance, it is not clear what we observe in Figure 5 leads to high diversity of sensitivity to different kinds of corruptions.

-= W3 =- The paper can benefit from explaining the experimental setup more explicitly, how many times do they repeat their experiments? What machines do they use? What are the hyperparmeters? etc.

**Summary Of The Paper:**

The paper evaluates NASBench-201 (NAS benchmark) on four adversarial attacks. First, they show that the architecture influences the robustness of the model, then they evaluate all of the NAS search space for various adversarial models. They show how this database can be used for various purposes, including training-free estimates of robustness and robustness-aware neural architecture search.


**Summary Of The Review:**

Well-motivated problem and a nice empirical analyses of an existing benchmark for a new set of metrics. This can be useful in furthering our understanding of network architecture design and robustness.

---

> ### Author Response · Authors · 2022-11-12
> **Reply**
>
> We appreciate the reviewer’s acknowledgement of the usefulness and novelty that our dataset provides and the insightful comments to improve our paper. We worked the reviewer’s suggestions into the new version of our draft:
> - [W1] We updated figure captions 2-7 with a sentence each to include their indications.
> - [W3] We added Section A.2 to our Appendix where we list all hyperparameter settings for adversarial attacks evaluated as well as the cluster specifications that evaluations were performed on.
>
> [W2] In regards to Figure 5: This Figure shows the Kendall ranking coefficient, which tends toward 0 between most corruption types. Hence, we conclude that different architecture designs perform more robustly towards different corruption types, which we describe as “high diversity of sensitivity to different kinds of corruptions” in the paper. Does this explanation address your concerns? If so, we are happy to update the description in the paper.
>
> We thank the reviewer again and look forward to further discussion.

---

> > ### Comment · Reviewer_VzZD · 2022-11-21
> > **Thank you!**
> >
> > I appreciate the clarification and the changes you have made to the paper. My rating is the same as before.

---

### Official Review · Reviewer_WniR · 2022-10-25

**Confidence:** 4
**Clarity, Quality, Novelty And Reproducibility:** see above
**Correctness:** 2
**Technical Novelty And Significance:** 3
**Empirical Novelty And Significance:** 2
**Recommendation:** 6

**Strength And Weaknesses:**

### Strength

* This paper presents a large-scale and systematic study of the relationship between robustness and architecture.
* The predictability of Jacobian and Hessian matrices is an important question. The results of this paper show their usefulness and limitation.

### Weaknesses

* The authors mentioned that "carefully crafting the topology of a network can have a substantial impact on its robustness, where networks with the same parameter count range in mean adversarial robust accuracy from 0.20% − 0.41%." I am not sure whether we can treat 0.2% - 0.41% adversarial accuracy as significant. In addition, what if we use a substantially different architect (but the same number of parameters) like Transformer?

**Summary Of The Paper:**

* This paper evaluates the adversarial robustness of 6466 non-isomorphic network designs from NAS-Bench-201.
* This paper benchmarks robustness measurements based on Jacobian and Hessian matrices for their robustness predictability.
* Neural architecture search on robust accuracies is performed to find the relationship between architecture and robustness under the same model size.

**Summary Of The Review:**

The conclusions reached by this paper are not well supported by quantitative results. Please correct me if I missed anything.

---

> ### Author Response · Authors · 2022-11-08
> **Reply**
>
> We appreciate the reviewer’s acknowledgement of the usefulness of our dataset and hope we can address all the concerns below.
>
> **1. “I am not sure whether we can treat 0.2% - 0.41% adversarial accuracy as significant”**
>
> The numbers in the abstract indeed contain a typo. It should read 0.2-0.41 (as in Section 4.3) or 20%-41%. We apologize for this error and correct it in our updated version. Our findings are therefore substantial.
>
> **2. “The conclusions reached by this paper are not well supported by quantitative results.”**
>
> Is this point addressed by correcting the above typo on the scale of the reached adversarial robustness? If not, could you provide a hint of which conclusions you are referring to?
>
> **3. “what if we use a substantially different architect [...] like Transformer?”**
>
> The motivation for our dataset is to provide an extensive evaluation of the robustness over small architectural design changes. In this sense, we agree that extending the dataset with Transformer models (via a search space that allows us to see the effect of small architectural changes) would be an interesting venture for future research and we thank the reviewer for this suggestion. The proposed design space covers commonly used CNN architectures and has been well argued for in [1,2], which collected 364 citations since 2020 according to google scholar.
>
> [1] Dong, Yang, NAS-Bench-201: Extending the Scope of Reproducible Neural Architecture Search, https://arxiv.org/abs/2001.00326
>
> [2]  Dong et al., NATS-Bench: Benchmarking NAS Algorithms for Architecture Topology and Size, https://arxiv.org/abs/2009.00437

---

> > ### Comment · Reviewer_WniR · 2022-11-15
> > **Thanks for the response**
> >
> > “The conclusions reached by this paper are not well supported by quantitative results.” Is this point addressed by correcting the above typo on the scale of the reached adversarial robustness? Yes. Thanks for your revision.
> >
> > In addition, I was wondering can model designer learn any intuition/lesson from the searching results from this paper in terms of robustness? For example, how do the size, order, and the type of layers impact the robustness of the network?

---

> > > ### Author Response · Authors · 2022-11-17
> > > **Reply**
> > >
> > > Dear Reviewer,
> > >
> > > Thank you for updating your recommendation. Are there any concerns left to also update your correctness and novelty scores?
> > >
> > > **"[...] can model designer learn any intuition/lesson from the searching results from this paper [...]?"**
> > >
> > > Yes, our manuscript provides an analysis of these aspects. We analyze in Section 4.3 (and Appendix B.2) cells with fixed parameter count and observe that stacking convolutional layers within a cell improves robustness. Additionally, we analyze in Appendix B.1 the edit distance of the best performing architectures based on (i) clean accuracy, (ii) mean adversarial accuracy, and (iii) mean corruption accuracy. Here we see that all those architectures are at least $2$ edits apart from each other, showing that promising architectures with respect to clean accuracies and robust architectures are significantly different from one another.
> > > Important to note here is that these are first observations, which can be made by using our provided dataset.
> > >
> > > Does this analysis motivate you to further increase your score?
> > >
> > > Thank you,
> > >
> > > Authors of Paper #2665

---

> ### Author Response · Authors · 2022-11-12
> **Reply**
>
> We thank the reviewer again for the input and hope we addressed all remaining concerns. We look forward to further questions and discussion.

---

### Author Response · Authors · 2022-11-12
**Changelog Revision 1**

# Changelog Revision 1
(changes in the PDF are blue)
- Abstract: There was a typo in the abstract we corrected. The last sentence should read "**20%-41%**" instead of "0.20%-0.41%".
- Appendix: We added Section A.2 about dataset gathering. This includes an algorithm and diagram that explains how we collected the data for our dataset as suggested by reviewer HhBB as well experiment settings as suggested by reviewer VzZD.
- Figure captions: We updated captions 2-7 to be more descriptive as suggested by reviewer VzZD.

---

### Decision · Program_Chairs · 2023-01-20

**Decision:**

Accept: poster

**Justification For Why Not Higher Score:**

The authors directly position their work over the existing dataset NAS-Bench-201 and further provide the adversarial robustness results through some adversarial attacks. The overall work could be a bit incremental, but all reviews agree on the usefulness of this augmented dataset in the field.

**Justification For Why Not Lower Score:**

Adversarial robustness is an important dimension that should be included in the existing NAS dataset. The authors also included interesting use cases to explore and exploit the proposed datasets, which may inspire the following further studies on the relevant datasets.

**Metareview: Summary, Strengths And Weaknesses:**

This paper enriched the NAS-Bench-201 dataset by including the adversarial robustness of 6466 non-isomorphic network designs. Though this is not the first time to show the connection between architecture and adversarial robustness, the dataset provided in this paper can be helpful for the following research in this direction.

**Note From Pc:**

if the above contains the word "oral" or "spotlight" please see: "oral" presentation means -> notable-top-5% and "spotlight" means -> notable-top-25%. As stated in our emails, we are disassociating presentation type from AC recommendations